# Production of probiotic garden cress (*Lepidium Sativum*) using *Bifidobacterium Bifidum* and its evaluation of nutritional value, biocontrol and growth rate ability

Golnaz Shambayati[1], Mojtaba Mohammadzadeh Vazifeh[1]*, Seyed Masoud Hosseini[2]*, Hasan Hajjami Barkousaraei[1]

**1** Faculty of Modern Biological Sciences and Technologies, University of Science and Culture, Tehran, Iran, **2** Faculty of Biological Sciences and New Technologies, SBU, Tehran, Iran

* mohammadzadeh.vazifeh@usc.ac.ir; Ma_Hosseini@sbu.ac.ir

## Abstract

Probiotics are one of the most beneficial elements in human health. Several studies have confirmed the health benefits of probiotics. The consumption of fermented vegetables is widespread worldwide and represents an important component of the human diet for a growing global population. In recent years, consumers have become more aware of the relationship between food and health, which has led to an increase in interest in functional foods. The global market for probiotic foods is growing rapidly due to the increasing consumer awareness. Although leafy vegetables are well-endowed with bacteria, including potentially probiotic strains of *Lactobacillus*, so far, plant-derived products have only been considered as carriers of probiotic cultures. This study aimed to produce probiotic garden cress using *Bifidobacterium bifidum* and to investigate its nutritional value, biocontrol, and growth rate and this is a great approach due to the increase in vegetarianism, lactose intolerance and people allergic to dairy products. *Bifidobacterium bifidum* (bb12) was inoculated into sterile cress seeds and the probiotic plant culture was done on days 0 and 3 after harvesting, in a TOS medium containing Mupirocin (MUP) antibiotic. The chemical properties (such as pH, acidity, and dry matter) of both samples were determined and the comparison of growth rate, fat content, vitamin C, and organoleptic properties was done. Bacterial survival in simulated gastrointestinal (SGI) conditions and its antifungal effect on *Rhizoctonia solani* were also analyzed. The number of bacteria on days 0 and 3 was $4 \times 10^{13}$ and $2.3 \times 10^{11}$ Colony Forming Unit/gram (CFU/g) respectively, which confirms the probiotic nature of the product (>$10^6$ bacteria). Also, significant difference ($P \leq 0.05$) was observed in the physicochemical parameters. Bacterial viability was reported as $6.3 \times 10^{10}$ CFU/g in simulated gastric condition and as $3 \times 10^6$ CFU/g in simulated intestine condition, and the antifungal effect of *Bifidobacterium* on *R. solani* was 15%. It can be concluded that production of other probiotic vegetables might

**Data availability statement:** All relevant data are within the paper and its Supporting Information files.

**Funding:** The author(s) received no specific funding for this work.

**Competing interests:** The authors have declared that no competing interests exist.

be possible and that the plant based probiotics can be used to support the growth of human intestinal bacteria and also maintain high cell viability during storage.

## Introduction

Foods which promote health beyond providing basic nutrition are termed "functional foods" [1]. The gut microbiota is mostly shaped by the environment, in particular the diet. Consumer behavior towards food choice is changing due to the profound understanding in the relationship between diet and health [2]. Nowadays, consumers are increasingly demanding products fortified with probiotic bacteria [3]. Probiotics are defined as living microorganisms that affect the health of the host by improving the composition of gut microflora. These microorganisms are a common ingredient in functional foods [4]. The intake of probiotics offers positive effects in preventing a range of diseases when consumed in adequate amount. It is commonly accepted that probiotics must be present at a minimum level, ranging from $10^6$_$10^9$ CFU/ml of the product, to be beneficial [5].

Probiotic viability in the food matrix depends on factors such as pH, storage temperature, oxygen levels, and the presence of competing microorganisms and inhibitors. Common microorganisms used in probiotic production are predominantly *Lactobacillus*, and *Bifidobacterium* species which produce organic acid and lower the intestinal pH [6]. Enterococci are lactic acid bacteria of importance in food, public health and medical microbiology [7]. Probiotic strains should be resistant to gastrointestinal conditions, and be safe for human use [8] and also being able to adhere to the intestinal epithelium to exert its beneficial effects. The adherence ability depends on the hydrophobicity and auto aggregation capacity of the probiotic microorganisms [9].

Some species from which specific strains have been isolated, characterized and proposed as probiotics include *Lactobacillus acidophilus*, *Lactobacillus gasseri*, *Lactobacillus johnsonii*, *Lactobacillus casei*, *Lactobacillus paracasei*, *Lactobacillus rhamnosus*, *Lactobacillus plantarum*, *Lactobacillus reuteri*, *Lactobacillus crispatus*, *Lactobacillus fermentum*, *Bifidobacterium animalis*, *Bifidobacterium bifidum.* There has been an increased interest during the last decade to add intestinal *Lactobacillus* spp. and *Bifidobacterium* spp. to fermented food products [38].

Most probiotic foods are milk-based but certain sectors of the population such as those allergic to milk proteins, those who are lactose intolerant, and those who are vegetarian, cannot consume dairy products. Therefore, a need has arisen among consumers for an alternative to fermented dairy products by exploring new non-dairy matrices as probiotics carriers [10]. Nondairy origin matrices are being increasingly used as supply vehicles for probiotics due to their high nutritional value in proteins, minerals, vitamins, fibers, antioxidants and other bioactive compounds. Furthermore, the lipid content found in some matrices of nondairy origin and the unique physiology of the plants help protect probiotics against different types of stressors [5]. Juices, fruits and vegetables enhance the survival of probiotics, as extra nutrients can be derived from the raw ingredients through cellular synthesis. Additionally, these nutrients serve as carriers for the incorporation of microorganisms due to the functional

benefits offered by various elements, including vitamins, minerals, antioxidants, and dietary fiber, which can render them excellent substrates for the growth of probiotics [11].

*Lepidium sativum*, also known as garden cress, is a fast-growing annual herbaceous plant belonging to the *Brassicaceae* family and can be grown at all elevations, all the year round [12]. The leaves of the plant are consumed raw in salads, also cooked with vegetable curries and used as garnish. The herb is an important medicinal plant since the Vedic era and is used in the treatment of cough and bleeding piles. Leaves are mildly stimulant, diuretic, and useful in scorbutic diseases and liver complaints [13].

The main aim of this research was to produce a probiotic garden cress by inoculating its seeds with *Bifidobacterium bifidum* (bb12) bacteria.

## Materials and methods

### Materials

Probiotic cells, *Bifidobacterium bifidium* (bb12), were provided by (Chr-Hansen A/S., Hørsholm, Denmark) and *Rhizoctonia. solani* fungi (IBRC-M 30523) was provided to determine the antifungal activity of *Bifidobacterium*. MRS Broth and MRS Agar mediums (Ibresco Co., Iran) and TOS Agar (Biomark, India) were used for culture activation and *Bifidobacterium* growth. PDA Agar (Merck, Kenilworth, NJ, USA) was used as the main medium for *Rhizoctonia. solani* culture. Hydrogen chloride (HCl), Sodium hydroxide (NaOH), and n-hexane solutions were purchased from (Dr. Mojallali Industrial Chemical Co., Tehran, Iran). Trypsin (Sigma Aldrich Co., St. Louis, MO, USA), Oxgall (Sigma Aldrich) and Pepsin (Sigma Aldrich) were utilized to prepare SGI conditions.

### Production of probiotic garden cress

**Culture revival.** The MRS broth medium was sterilized in an autoclave at $121°$C for 15 minutes. Bacteria were cultured in a sterile MRS broth medium and incubated anaerobically at $37°$C for 48–72 hours [14].

**Garden cress seed preparation. Surface sterilization**. One hundred grams of fresh garden cress seeds were purchased from the grocery store and divided equally into two sterile beakers. The seeds were surface sterilized in 70% ethanol for 1 minute, and hypochlorite solution for 10 minutes. Finally, the seeds were rinsed 5 times with sterile distilled water [15].

**Seed mucilage generation.** Each seed sample (fifty grams) was soaked separately in two containers (each containing 0.5 liters of sterile distilled water) for 12 hours [16].

**Seed inoculation.** The medium suspension containing *Bifidobacterium* strain was separated by centrifugation (Hettich Instruments, Tuttlingen, Germany) at 4500 rpm for 10 minutes at room temperature and after being washed twice with PBS buffer (Merck), the bacterial sediment was separated [17]. The inoculant was made at a ratio of 1:100 (inoculant: seed weight) by suspending bacterial colonies in sterilized ringer solution and making a solution with a turbidity of 0.5 McFarland ($1.5 \times 10^8$ CFU/ml) and a volume of 24 ml. Fifty grams of sterilized seeds were soaked in the inoculant containing *Bifidobacterium bifidum* for 2 hours [18].

### Seed planting

Six vases (three vases for each treatment) and cocopeat soil were prepared for planting seeds. Ten kg of soil was added to each vase [19]. Two liters of water were added to each vase to moisten the soil [20]. The plants were watered with 0.5 liters of water twice a day [21].

### Plant harvesting

Both treatments were harvested one week after planting when the plant height reached 5 cm. Both samples were separately transferred to the laboratory in sterile plastics [22].

 

## Comparison of duration and rate of growth

This was done by preparing a table and checking the height of the plant daily with a caliper.

## Bacterial enumeration

Bacterial enumeration was done according to the method described by Arepally and Goswami [14] with slight modifications. Five grams of the bacteria-containing sample were washed, cut, and dissolved in a 45 ml PBS solution and the serial dilution was performed. One ml of each dilution was cultured in TOS medium containing MUP (Merk) supplement by pour plate method and incubated in an anaerobic jar (Afshar Co., Iran) at $37°C$ for 72 hours. The viability of bacteria was examined on days 0 and 3 (due to the three-day expiration date of ready-to-eat vegetables). Samples were kept from day 0–3 at refrigerator temperature ($4°C$) in plastic bags.

## Physicochemical measurements

**pH.** The pH of both samples was measured according to organization for economic co-operation and development (OECD) guidelines for the testing of chemicals. After calibrating the pH meter (Crison Co., France), five grams of sample was mixed with 50 milliliters of distilled water, and the pH value was read at minutes one and two [23].

**Acidity.** The acidity of both samples was measured according to OECD guidelines for the testing of chemicals. To determine the acidity, five grams of each sample was titrated with 0.1N sodium hydroxide. The acidity of both samples was measured using the following formula:

$$\text{Acidity (calculated as linoleic acid)} = 1.31\, t \times c_1 / w(\%)$$

Where;

$$c_1 = (NaOH)\,.\,mol/L\ (normality)\ \text{of the solution}$$

$$t = \text{volume (mL) of NaOH solution}$$

$$w = \text{weight (g) of a sample} \tag{23}$$

## Dry matter percentage measurement

Dry matter was determined by drying the weighed samples in an oven (Shimaz, Iran) at $78°C$ for 24h. The following formula was used to measure the percentage of dry matter [24,25]:

$$\text{Percentage of dry matter} = \text{fresh sample weight-dry sample weight/fresh sample weight} \times 100$$

## Viability of bacteria under simulated gastrointestinal conditions

Simulated gastric juice was prepared by adding 0.35 g of pepsin in 100 ml of 0.2% saline and adjusting the pH to 2.0 and 2.5 with hydrochloric acid (for having the most similar condition to human SGI). Simulated intestinal juice was prepared by adding 0.1 g of trypsin and 1.8 g of bile salts in a sterile solution of 1.1 g of sodium bicarbonate and 0.2 g of sodium chloride in 100 ml distilled water and adjusting pH to 8.0 with 0.5 M sodium hydroxide. Both solutions were sterilized with filtering through 0.45 $\mu m$ filter [26].

## Total viable colony enumeration after exposure to SGI condition

The changes in the total number of living bacteria during the treatment with gastric juice (3 hours) and intestinal juice (3 hours) and they were determined by pour plate method using MRS agar on day 3 (due to the three-day expiration date of ready-to-eat vegetables). The plates were incubated anaerobically at $37°C$ for 48 hours. The number of *Bifidobacterium bifidum* in samples before and after the treatment was counted and compared. The viability of the bacteria was calculated according to the following equation:

$$\text{Survival rate (\%)} = \frac{\log \text{CFU N}_1}{\log \text{CFU N}_0} \times 100$$

Where $N_1$ represents the total viable count of *Bifidobacterium bifidum* after treatment by simulated gastrointestinal juices, and $N_0$ represents the total viable count of *Bifidobacterium bifidum* before treatment [26].

## Fat content

Fat content determination was done according to the method described by Stefano et al. [27] with slight modifications. Three grams of each sample were extracted in n-hexane solvent using a Soxhlet extractor for 6 hours. The extract was filtered and concentrated in a rotary evaporator at $40°C$. The amount of extracted oil was obtained using the following formula:

$$\text{Fat percentage = fat weight/sample weight} \times 100$$

## Vitamin C determination

The amount of vitamin C was determined according to ISO 6557/2–1984: Fruits, vegetables and derived products - Determination of ascorbic acid - Part 2: Routine methods [28]. The amount of vitamin C was measured by extracting ascorbic acid from the samples, using the oxalic acid (Sigma Aldrich) solution with the acetic acid, and titration with 2,6-dichlorophenol-indophenol (Merk) until a light pink color appeared. Vitamin C content (mg) was calculated according to the following equation:

$$\text{Vitamin C} = \frac{(V_0 - V_1) \times m_1}{m_0} \times 100$$

Where $V_0$ represents the volume of dye solution (ml) used for titration, $V_1$ represents the volume of the dye solution (ml) used in the control test, $m_1$ represents the mass of ascorbic acid (mg) and $m_0$ represents the mass of the sample (g) used in the titration test.

## Sensory evaluation

The characteristics of Garden cress in terms of taste (spiciness), aroma, texture, color, and appearance were examined by five panelists. The organoleptic evaluation in this research was based on the hedonic sensory evaluation, which is a 5-point method (from most unfavorable to most favorable). Testing was carried out by giving a questionnaire to the panelists with parameters, i.e., color, aroma, taste, texture, and appearance. To achieve quantitative results, the qualitative results obtained from the test were converted into quantitative (parametric) data and results by giving the points from 1 to 5 to properties [29].

## Antifungal activity of *Bifidobacterium bifidum*

The agar well diffusion method was used to analyze the antifungal activity of bacteria. To prepare the supernatant, *Bifidobacterium bifidum* bacteria were cultured on an MRS broth medium for 48 hours. Then it was centrifuged at 3000 rpm for 15 minutes. One ml of the supernatant was filtered by a 0.45 $\mu m$ syringe filter. On the other hand, a plate containing sterile PDA (Potato Dextrose Agar) medium was prepared and 4 wells with a diameter of 0.6 mm were bored using a sterile punch. 100 $\mu l$ of the supernatant was added to the wells (until completely filled) and one well was considered as a control sample containing sterile distilled water. An agar plug (0.5 × 0.5 cm) was removed from the fresh culture of *Rhizoctonia solani*, and placed in the center of the plate and between the wells. The plate was incubated for 4 days at 25°C. The radius of the growth zone (cm) was measured in both samples, and the results were recorded after 4 days. Antifungal activity was calculated according to the following formula:

$$FI\,(\%) \;=\; \frac{R_c - R_t}{R_c} \times 100$$

Where $R_c$ represents the radius of the growth zone in the control and $R_t$ represents the radius of the growth zone in *Bifidobacterium*-containing treatment [30].

## Statistics

All experimental results are mean $\pm$ S.D. of three parallel experiments. One-way analysis of variance (ANOVA) was used to compare the groups. Differences were considered significant at $P \leq 0.05$.

## Results

### Comparison of duration and rate of growth

The result of this measurement and comparison of the data showed that the addition of *Bifidobacterium* to the plant did not change the rate and time of its growth (Fig 1), and both treatments had equal growth under the same planting conditions in terms of light and watering (Fig 2).

### Probiotic count

The number of *bifidobacteria* in the garden cress was determined as the average of three counts (Fig 3). According to recommendations, there should be a minimum of $10^6$ CFU/ml or g of live probiotic bacteria at the time of consumption. Counting *bifidobacteria* in inoculated garden cress shows that plant is indeed a probiotic vegetable.

### Physicochemical properties analysis

The pH value of the control sample and the *bifidobacteria* – containing one was determined at minutes one and two. The difference in the pH of the two treatments was significant ($P \leq 0.05$) and the low pH of the probiotic sample can confirm the presence of *Bifidobacterium* (which produces acid) in the garden cress (Fig 4).

To determine the acidity, 5 grams of each sample was measured, and individually titrated with 0.1N sodium hydroxide. The acidity of each sample was calculated according to the formula. The results show the treatment sample had a significant ($P \leq 0.05$) lower acidity (Fig 5):

$$Control = 0.01233 \pm 0.002\%$$

$$Probiotic = 0.01833 \pm 0.003\%$$

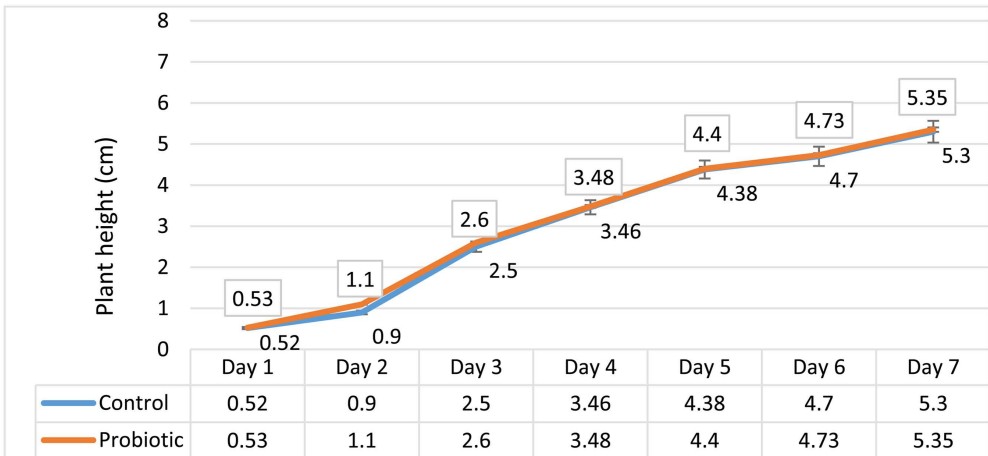

**Fig 1. Growth rate comparison of the probiotic and control samples.**

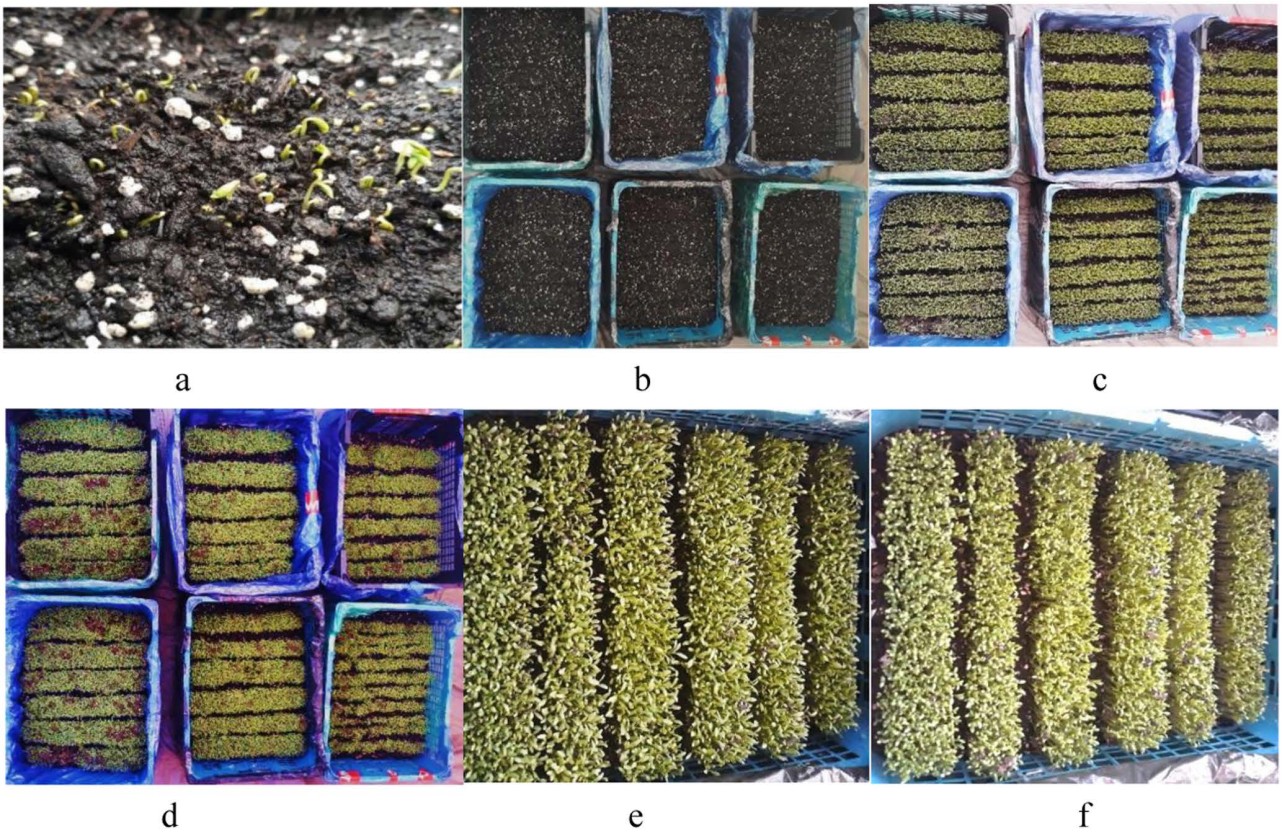

**Fig 2. Plant growth process from day 1 to 6. a: day 1 (24 hours after sowing).** Seed germination. The average height of bifidobacterium – containing plant and the control one was respectively, 0.53 cm and 0.52 cm. b: day 2. The above vases show the bifidobacterium – containig plants and the below ones show the control plants. The average height of probiotic and the control plant on this day was 1.1 cm 0.9 cm respectively. c: day 3. Probiotic plant (2.6 cm), control plant (2.5 cm). d: day 4. Probiotic plant (3.48 cm), control plant (3.46 cm). e: day 5. Probiotc plant (4.4 cm), control plant (4.38 cm). f: day 6. Probiotic plant (4.73 cm), control plant (4.7 cm).

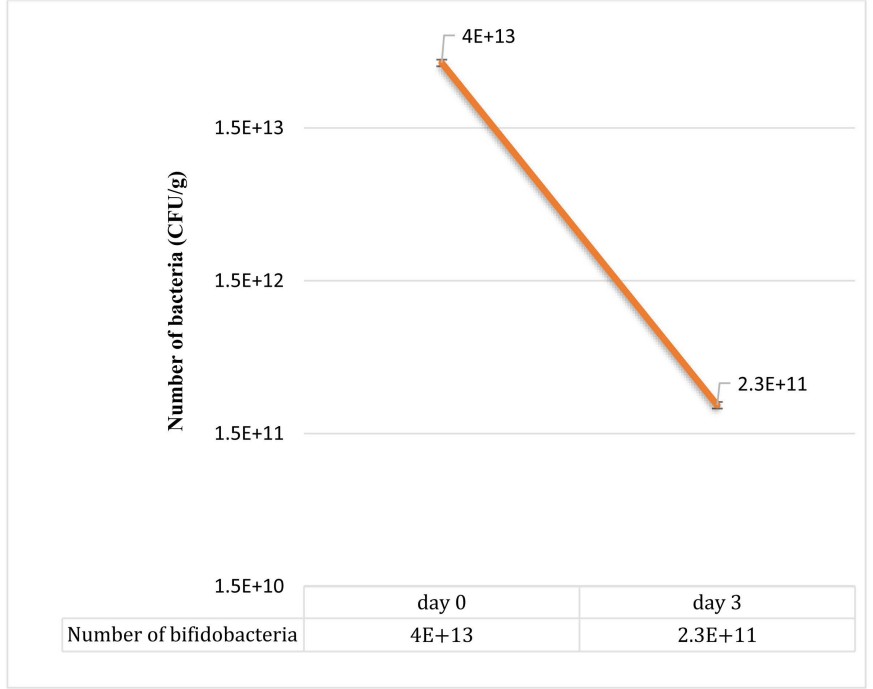

**Fig 3. Changes in the number of Bifidobacteria from days 0 to 3.** A decrease in the bacterial cells number is observed but the product maintained its probiotic properties.

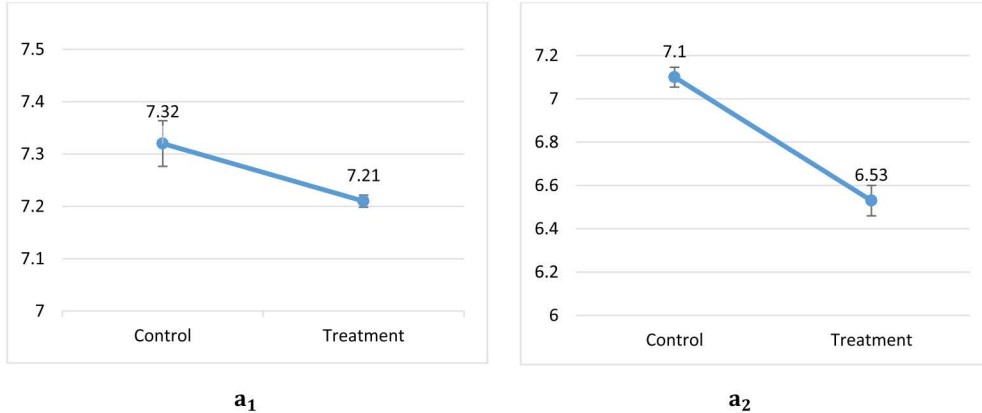

**Fig 4. pH values of two samples at minutes 1 ($a_1$) and 2 ($a_2$).**

## Dry matter

The weight of the samples before and after being placed in the oven was measured and the percentage of dry matter was calculated according to the below formula:

$$Control\ sample = 94.0500\% \pm 0.0361\%$$

$$Probiotic\ sample = = 94.2400\% \pm 0.1353\%$$

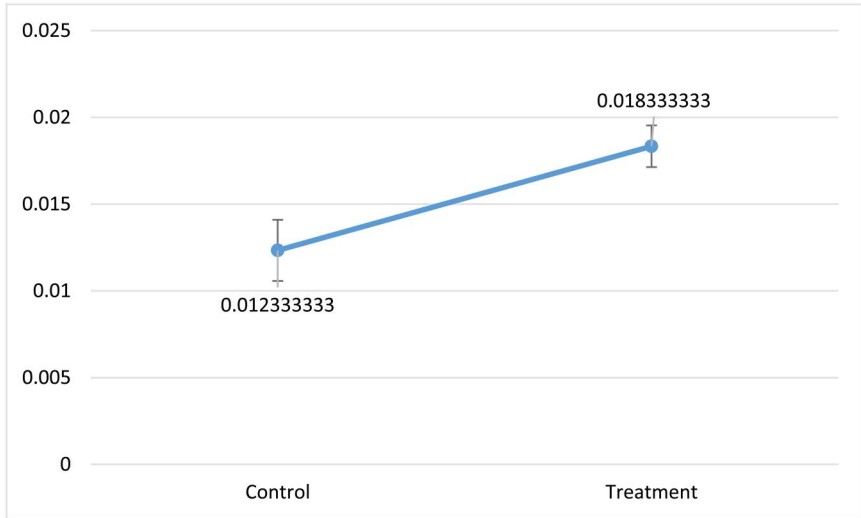

**Fig 5. The acidity difference of two samples; The samples containing bacteria represents higher acidity.**

The results showed no significant differences ($P \geq 0.05$)

### The viability of *bifidobacteria* under gastrointestinal conditions

The number of viable *bifidobacteria* in the garden cress reached $6.3 \times 10^{11} \pm 1.15E+11$ CFU/g after exposure to the simulated gastric condition. The number of viable *bifidobacteria* was $3 \times 10^{6} \pm 1.3E+6$ under simulated intestinal conditions. So, we conclude that the resulting product could survive in the simulated gastrointestinal conditions, which is a significant point because one of the points in the production of probiotic foods is their survival in the human digestive system and resistance to gastric and intestinal conditions.

### Fat content

The amount of fat (%) was determined by measuring the mass of the samples and the Soxhlet extraction apparatus before and after being placed in a Soxhlet extractor and rotary evaporator.

The results of this measurement show that the presence of *Bifidobacterium* significantly ($P \leq 0.05$) increased the fat content of garden cress (Fig 6).

### Vitamin C content

The volume of dye solution used for titration was 0.2 ml, the volume of the solution used in the control test was 0.1 ml, the mass of the ascorbic acid was 50 mg, and the mass of the samples used for the titration was approximately 5 grams (Fig 7). The results show significant ($P \leq 0.05$) difference.

### Organoleptic properties analysis

Three groups (each included five panelists) evaluated organoleptic properties of both control and treatment samples, the results of which are shown as the average of each parameter (from 1 to 5) in the graph below (Fig 8). The results of none of the parameters examined were significant ($P \geq 0.05$):

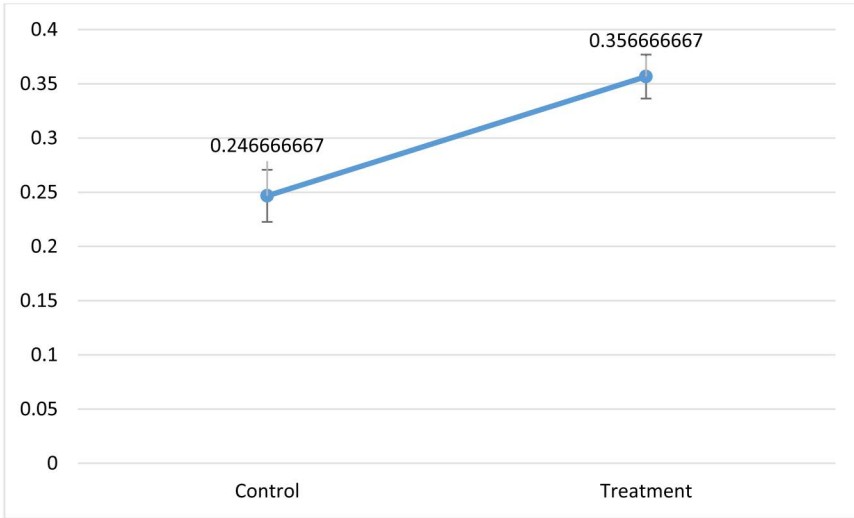

**Fig 6. Changes in fat content of two samples.**

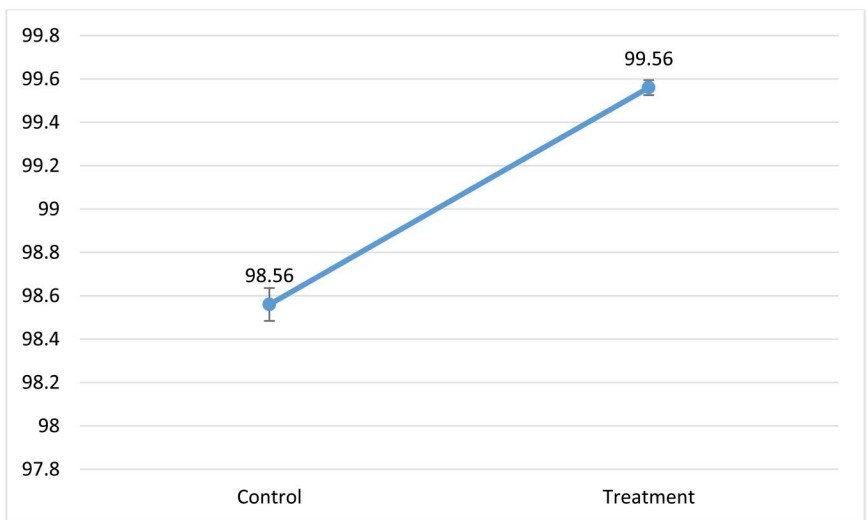

**Fig 7. Vitamin C content of the probiotic and the control samples.**

### Antifungal activity of *Bifidobacterium bifidum* (bb12)

The radius of the growth zone in the control sample was reported as 3.8 cm, and in the sample containing *Bifidobacterium*, it was reported as 3.2 cm. The percentage of fungi growth inhibition by *Bifidobacterium* was calculated according to the following formula:

$$FI = \frac{3.8 - 3.2}{3.8} \times 100 = 15\%$$

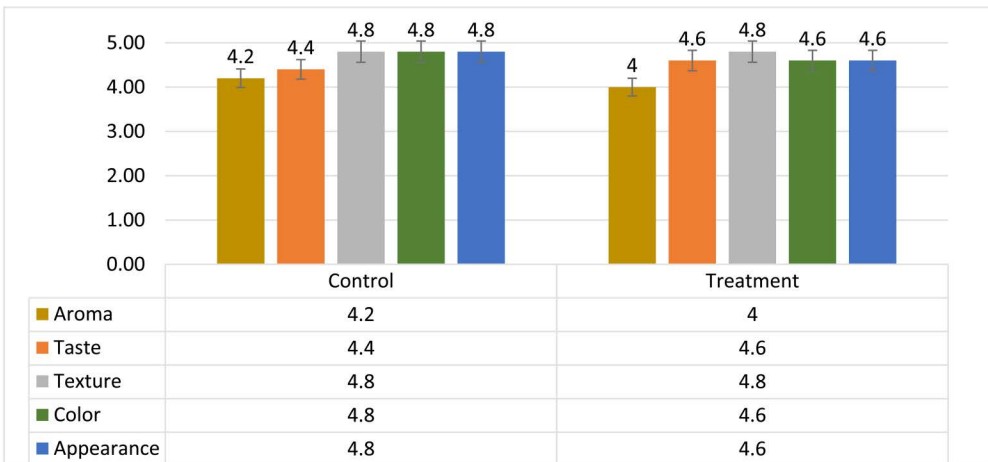

**Fig 8. Sensory evaluation results as an average of fifteen determinations.**

## Discussion

In confirmation of the obtained results in the enumeration and survival of bacteria and the decrease in the number of them during the storage period, Babak Pakbin et al. [31] observed that the number of *Lactobacillus delbruecki* bacteria in their product reached approximately $10 \times 10^9$ CFU/ml and it was suitable for consumption as a probiotic drink. A decrease in the number of bacteria was observed from weeks 0–4, but there was no change in the probiotic properties of the product. Tatiana Colombo Pimentel et al. [32] investigated the probiotic viability of apple juice supplemented with two probiotic strains and reported a rapid loss in the number of bacteria during the storage of juices. It has been reported by Min et al. [33] that the food matrix used to inoculate probiotic strains can affect their viability and announced that the population of the three strains used, reduced with increasing storage time. Ankita Kataria et al. [34] reported that it is possible to deliver *Bifidobacterium longum* through ice cream. The legume sprouts and *Lb. plantarum* 299V can be successfully combined to form a new probiotic-rich food [35]. The enumeration of the bacteria in the present study showed the essential count for probiotic bacteria to exhibit beneficial effects.

We observed a lower pH and higher acidity in the probiotic sample in comparison with the control one, which confirms the presence of *Bifidobacterium* and its acid production but no changes in dry matter were reported. Oliveira et al. [36] reported an increase in dry matter of goat cheese and stated that this could be related to an increase in the percentage of protein. So, we can conclude that changes in dry matter might be observed in dairy probiotic products. Vinderola et al. [37] reported that the pH values in two types of yogurt containing *Bifidobacterium* and *Lactobacillus* were negligible during the refrigerated storage. The garden cress pH in the present study was in the range of 6.5–7.2 which includes the optimum pH for the growth of *Bifidobacterium*. Many researches have confirmed that the presence of probiotic strains in products does not have a significant effect on the titratable acidity or may increase it [38]. The decrease in pH value in honey yogurt containing *Lactobacillus reuteri* DPC16 during the 3 weeks of storage was also reported by Anand Mohan et al. [39].

The survival of probiotic strains is an important factor for exertion their health benefits. In present study, the viability of *Bifidobacterium* decreased after exposure to SGI condition but still was higher than the minimal range accepted to be beneficial ($\times 10^6$ CFU/ml or g). This result is similar to those reported by Casarotti et al. [40] who demonstrated that the viability of probiotic strains significantly decreased after exposure to in vitro gastrointestinal conditions. Moreover, Chávarri et al. [41] observed a rapid loss in the number of free bacterial cells in SGI conditions. In a more recent study by Khorasani and Shojaosadati [42] it has been found that the number of viable probiotic bacteria decreased during exposure to SGI conditions.

It has been demonstrated by many studies that the fat content of foods provides protection for probiotic bacteria. In the present study, the fat content of probiotic sample was not significantly higher than the control one. Ciesielska et al. [21] have shown that the kind of water is what affects the fat content in garden cress. Popova [43] reported that the addition of probiotic strains reduces the saturated fatty acids and increases the polyunsaturated fatty acids of meat, and thus has a positive effect on it.

We also observed that the Vitamin C content was a bit affected by *Bifidobacterium* and its value difference in both samples was 1mg (higher in probiotic sample). AdebayoTayo et al. [29] found that the vitamin C content in probiotic pineapple juice had no significant differences during storage but no significant change in Vitamin C content of the fermented Acerolla ice cream was observed by Favaro Tritade et al. [44]. Flores-Félix et al. [45] reported that the vitamin C highly increased in the strawberries inoculated with plant probiotics.

The fungal growth inhibition of *Bifidobacterium* against *R. solani* that we observed in present study was 15%. Xin Ma et al. [46] also worked on another probiotic strain, *Bacillus subtilis*, and reported that *B. subtilis* has been used widely in agriculture as a biocontrol agent and has a high antifungal activity against *R. solani*. Gerbaldo et al. [47] have also indicated the antifungal effect of two *Lactobacillus* strains against aflatoxigenic fungal isolate.

The organoleptic evaluation showed no difference in *Bifidobacterium* – containing garden cress. Coman et al. [1] worked on four probiotic foods, and reported that no significant differences were observed between probiotic-enriched foods and control foods. Afzaal et al. [48] demonstrated that the taste and the flavor of probiotic and control ice cream were the same and no difference was observed.

## Conclusion

Dairy products have traditionally been considered as the best carrier for probiotic bacteria. Allergy to dairy products, lactose intolerance, and cholesterol-containing substances are major disadvantages associated with the use of fermented dairy products for a large percentage of consumers. For this reason, probiotic products derived from fermentation of cereals, fruits, and vegetables have attracted the attention of the scientific world as well as consumers.

The potential of vegetables as carriers of probiotics is increasing since these foods are rich in carbohydrates, vitamins, and minerals. In addition, vegetables are rich in phytochemicals and nutrients, and their physical structure provides a protective environment for probiotics. Fermented or minimally processed vegetables are alternatives to plant-based products and have been considered as carriers of probiotics, including: beets, carrots, radishes, artichokes, cabbage, broccoli, celery, aloe vera, soybeans, almonds, and walnuts. According to mentions above, production of probiotic vegetables is of great value and might help people suffering from diseases related to the consume of dairy products and also the vegan population refusing consumption of animal products.

In this study, we produced probiotic garden cress by inoculating *Bifidobacterium bifidum* (bb12) to its seed. Most importantly, the bacteria could resist in simulated gastrointestinal condition and the garden cress retained its probiotic properties during the 3 days of storage. The bacterial existence in the plant tissue did not reduced its nutritional value such as vitamin C and fat content. The survival of bacteria in the garden cress confirmed that vegetables might be effective carriers for probiotics and ensure their survival in harmful conditions of the digestive tract. This is a great achievement due to the increase in lactose intolerant population and vegetarianism and thus increasing demand for nondairy probiotic foods. This work shows that non-dairy, vegetable-based products are suitable substrates and can support high cell viability during storage.

In conclusion, inoculation of other probiotic strains to other types of edible vegetables might be a successful method to produce probiotic vegetable. Although there is a large quantity of traditional fermented food produced from different substrates but enrichment of vegetables with probiotics is likely to be a successful attempt and yields products for a new branch of functional foods.

## Supporting information

**S1 Table.  Bacteria count (A), means (B) and analysis of variance (C).**
(PDF)

**S2 Table.  Acidity Measurement (A), means (B) and analysis of variance (C).**
(PDF)

**S3 Table.  pH measurement of control and treatment samples (A), Means (B) and analysis of variance at minute one.**
(PDF)

**S4 Table.  pH measurement of control and treatment sample (A), means (B) and analysis of variance (C) at minute two.**
(PDF)

**S5 Table.  Dry matter measurement (A), means (B) and analysis of variance (C).**
(PDF)

**S6 Table.  Fat content measurement (A), means (B) and analysis of variance (C).**
(PDF)

**S7 Table.  Colony count after gastrointestinal juice exposure (A), means of intestinal exposure results (B), means of colony count before gastrointestinal exposure (C), means of gastric juice exposure (D).**
(PDF)

**S8 Table.  Vitamin C content measurement (A), means (B), and analysis of variance (C).**
(PDF)

**S9 Table.  Organoleptic properties scores of 15 panelists.**
(PDF)

**S10 Table.  Aroma Analysis of variance (A) and means (B).**
(PDF)

**S11 Table.  Taste Analysis of Variance (A) and means (B).**
(PDF)

**S12 Table.  Texture analysis of variance (A) and means (B).**
(PDF)

**S13 Table.  Color analysis of variance (A) and means (B).**
(PDF)

**S14 Table.  Appearance analysis of variance (A) and means (B).**
(PDF)

## Author contributions

**Conceptualization:** Golnaz Shambayati, Mojtaba Mohammadzadeh Vazifeh.

**Investigation:** Golnaz Shambayati, Hasan Hajjami Barkousaraei.

**Methodology:** Golnaz Shambayati, Mojtaba Mohammadzadeh Vazifeh, Hasan Hajjami Barkousaraei.

**Project administration:** Mojtaba Mohammadzadeh Vazifeh, Seyed Masoud Hosseini.

**Resources:** Mojtaba Mohammadzadeh Vazifeh.

**Validation:** Mojtaba Mohammadzadeh Vazifeh, Seyed Masoud Hosseini.

**Writing – original draft:** Golnaz Shambayati.

**Writing – review & editing:** Mojtaba Mohammadzadeh Vazifeh.

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
