## [Decision Letter · Decision Letter 0]

24 Jun 2024

Dear Dr. Shambayati,

Thank you for submitting your manuscript to PLOS ONE. After careful consideration, we feel that it has merit but does not fully meet PLOS ONE’s publication criteria as it currently stands. Therefore, we invite you to submit a revised version of the manuscript that addresses the points raised during the review process.

We look forward to receiving your revised manuscript.

Kind regards,

Khaled Abd EL-Hamid Abd EL-Razik, Ph.D.

Academic Editor

PLOS ONE

Journal Requirements:

2. In the online submission form, you indicated that The data underlying the results presented in the study are available from (Golnaz Shambayati/gshambayati97@gmail.com).

Reviewers' comments:

Reviewer's Responses to Questions

**Comments to the Author**

1. Is the manuscript technically sound, and do the data support the conclusions?

Reviewer #1: Partly

Reviewer #2: Yes

Reviewer #3: Yes

2. Has the statistical analysis been performed appropriately and rigorously?

Reviewer #1: N/A

Reviewer #2: No

Reviewer #3: N/A

3. Have the authors made all data underlying the findings in their manuscript fully available?

Reviewer #1: Yes

Reviewer #2: Yes

Reviewer #3: Yes

4. Is the manuscript presented in an intelligible fashion and written in standard English?

Reviewer #1: Yes

Reviewer #2: Yes

Reviewer #3: No

**Reviewer #1:**  please follow instruction within the manuscript . authors should add the impact of research in the conclusion and introduction need more improvements to be more informative

some spelling mistakes need to be corrected

add more details in materials section as probiotic inoculation

**Reviewer #2:**  the manuscript is techinically sound and have an impact for future applying also the scientific quality of the paper is good,Statistical Analysis is required.the authors have made all data underlying the findings in the manuscript fully available, the language of the manuscript is clear .Here are some comments for revising it.

I have read the manuscript entitled

‘Production of probiotic garden cress (Lepidium Sativum) using Bifidobacterium

Bifidum and its evaluation of nutritional value, bio control and growth rate’ thoroughly. The manuscript is discussing an interesting topic; generation of probiotic garden seeds and assessment of bifidobacteria's viability and resistance in gastrointestinal environments in addition, analyses of its organoleptic qualities, growth rate, and effect on vitamin C content were conducted.

• Where is the key word?

• SGI must be written fully for the first time in abstract.

• Line 93 SGI can be written in abbreviated form as it is mentioned before.

• Line 123 correct the word moistens.

• Line 175 they were determined instead of and were determined.

• Why the authors didn’t make different intervals in determination of total viable count for studying the effect of stomach condition, also why they didn’t make different PH values?

• Why the authors didn’t make different intervals in determination of total viable count for studying the effect of SGI with different concentration for bile condition?

• Line 220 write the name of PDA media (Potato Dextrose Agar) media fully.

• Statistical Analysis is required.

• Conclusion: The whole conclusion should be summarized and include recommendation with mentioning the most important finding not to repeat all the result that presented in detail in the result section.

**Reviewer #3: ** The manuscript needs English editing, reconstitution of sentences

follow Reviewer comments

No statistical analysis was performed

follow journal guidelines

Have the authors made all data underlying the findings in their manuscript fully available?

The authors make all data underlying the findings described in the manuscript fully available without restriction

**Do you want your identity to be public for this peer review?** For information about this choice, including consent withdrawal, please see our Privacy Policy

Reviewer #1: **Yes: ** Amany Ahmed Arafa

Reviewer #2: **Yes: ** Eman Shafeek Ibrahim

Reviewer #3: **Yes: ** Ashraf S. Hakim

---

## [Author Response · Author response to Decision Letter 1]

2 Dec 2024

Dear PLOS one team

I hope I’ve been able to amend and complete the article correctly. This is my first experience submitting an article and I was not aware of some basis needed to do so.

*Response to reviewer #1

- Please follow instruction within the manuscript. authors should add the impact of research in the conclusion and introduction need more improvements to be more informative.

The instructions have been followed and the impact of the research has been added. The introduction and the conclusion sections have been restructured and more informative.

- Some spelling mistakes need to be corrected

All the spelling and numerical mistakes have been corrected.

- Add more details in materials section as probiotic inoculation

All of the materials and methods have completely been added and the probiotic inoculation method has also been rewritten and fully explained.

*Response to reviewer #2

- Where is the key word?

The key words are added to the end of the abstract section

- SGI must be written fully for the first time in abstract.

SGI is fully rewritten in the abstract.

- Line 93 SGI can be written in abbreviated form as it is mentioned before.

- Line 123 correct the word moistens.

- Line 175 they were determined instead of and were determined.

All the spelling, abbreviation and grammatical mistakes have been corrected.

- Why the authors didn’t make different intervals in determination of total viable count for studying the effect of stomach condition, also why they didn’t make different PH values?

- Why the authors didn’t make different intervals in determination of total viable count for studying the effect of SGI with different concentration for bile condition?

The strain used in this study (Bifidobacterium bifidum) is an industrial strain whose viability and resistance to gastrointestinal condition and pH has already been examined and confirmed. We aimed to investigate the ability of the bacteria to survive in stomach and intestinal conditions due to the change in bacterial niche (The original niche of this strain is the human large intestine not the herbs). For this reason, only the pH mentioned in the references and articles discussing probiotics viability was used to check the survival of the bacteria and there was no need to use different concentrations of bile salts or other pH values.

Determination of total viable count after exposure to SGI condition was done on day 3 because it was important for us that the bacteria survive in the mentioned conditions until the last day of expiration. That is why different intervals was not made.

- Line 220 write the name of PDA media (Potato Dextrose Agar) media fully.

The full form of PDA media is written.

- Statistical Analysis is required.

Some tests including fat content, pH and acidity measurement was repeated two more times with the mentioned protocols to add statistical analysis you asked for. The other parameters, except organoleptic evaluation, are reported as an average of three results obtained from previous experiments but the statistical analysis is also added to them, hoping that it will be acceptable to you.

- Conclusion: The whole conclusion should be summarized and include recommendation with mentioning the most important finding not to repeat all the result that presented in detail in the result section.

The most important findings and recommendations are added to the conclusion section and the results repetition is removed.

*Response to reviewer #3

- The manuscript needs English editing, reconstitution of sentences

English editing and reconstitution of sentences are done. I tried to rewrite the manuscript in an intelligible fashion and standard English.

- Follow Reviewer comments

All the comments have been applied and followed.

- No statistical analysis was performed

The statistical analysis is performed and added.

- Follow journal guidelines

The journal guidelines have been followed.

- Have the authors made all data underlying the findings in their manuscript fully available?

Yes; The authors made all data underlying the findings in the manuscript fully available without restriction.

All the mistakes the dear reviewers have mentioned were completely correct and acceptable and I am very grateful that you read my article carefully.

Thank you for giving me another chance to send you the revised manuscript. I hope I can send you more articles and have more cooperation with you.

Kind regards

Golnaz Shambayati

---

## [Decision Letter · Decision Letter 1]

30 Jan 2025

Dear Dr. Shambayati,

Thank you for submitting your manuscript to PLOS ONE. After careful consideration, we feel that it has merit but does not fully meet PLOS ONE’s publication criteria as it currently stands. Therefore, we invite you to submit a revised version of the manuscript that addresses the points raised during the review process.

We look forward to receiving your revised manuscript.

Kind regards,

Shengqian Sun

Academic Editor

PLOS ONE

Journal Requirements:

Reviewers' comments:

Reviewer's Responses to Questions

**Comments to the Author**

Reviewer #2: All comments have been addressed

Reviewer #4: All comments have been addressed

2. Is the manuscript technically sound, and do the data support the conclusions?

Reviewer #2: Yes

Reviewer #4: Yes

3. Has the statistical analysis been performed appropriately and rigorously?

Reviewer #2: Yes

Reviewer #4: Yes

4. Have the authors made all data underlying the findings in their manuscript fully available?

Reviewer #2: Yes

Reviewer #4: Yes

5. Is the manuscript presented in an intelligible fashion and written in standard English?

Reviewer #2: Yes

Reviewer #4: Yes

Reviewer #2: The authors have carried out significant changes to the manuscript. They have addressed most of the suggested corrections and comments. Really, it's an interesting study that has a significant impact. Now, the manuscript could be accepted.

Congratulations.

Reviewer #4: The research in this paper is very interesting and the results are also very meaningful. The production of probiotics using Bifidobacterium bifidum is of great research value. While the writing needs to be more standardized, it is recommended to make appropriate revisions.

1.The Introduction is well written, but the length is a bit short. Please add some content and references appropriately.

2.The vertical axis of Figure 4 needs to be explained.

3.There are too many images in the manuscript, so it is necessary to consider combining them into larger images, or some images can be included in the appendix. Especially the merging of Figures 11-13.

4.Similarly, the results section is also appropriately merged together based on the merging of the graphs.

5.The conciseness of the research conclusions is not enough, for example, the conclusion and abstract sections do not have a research significance section, only the research results.

**Do you want your identity to be public for this peer review?** For information about this choice, including consent withdrawal, please see our Privacy Policy

Reviewer #2: **Yes**

Reviewer #4: No

---

## [Author Response · Author response to Decision Letter 2]

9 Feb 2025

Dear Reviewer 4

I hope you are doing great. Thank you so much for analyzing my manuscript carefully. The points you mentioned are all acceptable and I'm so grateful to announce that all of them have been revised and corrected.

I hope I get acceptance letter

Best regards

Golnaz Shambayati

---

## [Editor Report · Decision Letter 2]

11 Feb 2025

Dear Dr. Shambayati,

Thank you for your revised manuscript. I appreciate your efforts in addressing the reviewers' comments and completing the revision. However, I noticed that the submission is missing a properly marked-up version with track changes as well as the unmarked version of the revised manuscript. 

**A rebuttal letter that responds to each point raised by the academic editor and reviewer(s). You should upload this letter as a separate file labeled 'Response to Reviewers'.****A marked-up copy of your manuscript that highlights changes made to the original version. You should upload this as a separate file labeled 'Revised Manuscript with Track Changes'.****An unmarked version of your revised paper without tracked changes. You should upload this as a separate file labeled 'Manuscript'.**

Please carefully follow the journal's submission guidelines and consider referring to examples from fellow researchers to ensure accuracy. Kindly revise and resubmit accordingly.

Let me know if you need any further clarification.

Kind regards,

Shengqian Sun

Academic Editor

PLOS ONE
---

## [Author Response · Author response to Decision Letter 3]

22 Mar 2025

Dear PLOS one team

I hope I’ve been able to amend and complete the article correctly. This is my first experience submitting an article and I was not aware of some basis needed to do so.

*Response to reviewer #1

- Please follow instruction within the manuscript. authors should add the impact of research in the conclusion and introduction need more improvements to be more informative.

The instructions have been followed and the impact of the research has been added. The introduction and the conclusion sections have been restructured and more informative.

- Some spelling mistakes need to be corrected

All the spelling and numerical mistakes have been corrected.

- Add more details in materials section as probiotic inoculation

All of the materials and methods have completely been added and the probiotic inoculation method has also been rewritten and fully explained.

*Response to reviewer #2

- Where is the key word?

The key words are added to the end of the abstract section

- SGI must be written fully for the first time in abstract.

SGI is fully rewritten in the abstract.

- Line 93 SGI can be written in abbreviated form as it is mentioned before.

- Line 123 correct the word moistens.

- Line 175 they were determined instead of and were determined.

All the spelling, abbreviation and grammatical mistakes have been corrected.

- Why the authors didn’t make different intervals in determination of total viable count for studying the effect of stomach condition, also why they didn’t make different PH values?

- Why the authors didn’t make different intervals in determination of total viable count for studying the effect of SGI with different concentration for bile condition?

The strain used in this study (Bifidobacterium bifidum) is an industrial strain whose viability and resistance to gastrointestinal condition and pH has already been examined and confirmed. We aimed to investigate the ability of the bacteria to survive in stomach and intestinal conditions due to the change in bacterial niche (The original niche of this strain is the human large intestine not the herbs). For this reason, only the pH mentioned in the references and articles discussing probiotics viability was used to check the survival of the bacteria and there was no need to use different concentrations of bile salts or other pH values.

Determination of total viable count after exposure to SGI condition was done on day 3 because it was important for us that the bacteria survive in the mentioned conditions until the last day of expiration. That is why different intervals was not made.

- Line 220 write the name of PDA media (Potato Dextrose Agar) media fully.

The full form of PDA media is written.

- Statistical Analysis is required.

Some tests including fat content, pH and acidity measurement was repeated two more times with the mentioned protocols in order to add statistical analysis you asked for. The other parameters, except organoleptic evaluation, are reported as an average of three results obtained from previous experiments but the statistical analysis is also added to them, hoping that it will be acceptable to you.

- Conclusion: The whole conclusion should be summarized and include recommendation with mentioning the most important finding not to repeat all the result that presented in detail in the result section.

The most important findings and recommendations are added to the conclusion section and the results repetition is removed.

*Response to reviewer #3

- The manuscript needs English editing, reconstitution of sentences

English editing and reconstitution of sentences are done. I tried to rewrite the manuscript in an intelligible fashion and standard English.

- Follow Reviewer comments

All the comments have been applied and followed.

- No statistical analysis was performed

The statistical analysis is performed and added.

- Follow journal guidelines

The journal guidelines have been followed.

- Have the authors made all data underlying the findings in their manuscript fully available?

Yes; The authors made all data underlying the findings in the manuscript fully available without restriction.

*Response to reviewer #4:

- The Introduction is well written, but the length is a bit short. Please add some content and references appropriately

The introduction is written longer and the content related to this section is added properly.

- The vertical axis of Figure 4 needs to be explained.

Done!

- There are too many images in the manuscript, so it is necessary to consider combining them into larger images, or some images can be included in the appendix. Especially the merging of Figures 11-13.

You are completely right. I felt the same when looking at the file. Some unnecessary figures have been eliminated and the figures and tables labels are also changed due to the rearranges in the figures and tables order.

- Similarly, the results section is also appropriately merged together based on the merging of the graphs.

Done!

- The conciseness of the research conclusions is not enough, for example, the conclusion and abstract sections do not have a research significance section, only the research results

No doubt, the points you mentioned are definitely right. Thank you for your points of view. I should have mentioned the importance of my research and highlight it in order the readers know WHY I decided to go work on this topic. I tried my best to show its importance in the conclusion and abstract section. I hope I was successful.

The ‘Response to reviewers’ and ‘Manuscript’ files have already been written and prepared. But unfortunately I had a problem preparing ‘Revised Manuscript with Track Changes’ file. I did not get your mean by “Track Changes” and every time you mentioned a mistake and asked for a revision, I highlighted the edited section to let you know that this part have been edited, eliminated or added to the manuscript. Actually I have included the changes in the highlighted sections of the document. I am so sorry for that. I have been dealing with many problems lately and it completely confused me. It would be a great favor if you accept this version of my manuscript. Thank you. I’ll be sending you more papers soon and I promise I will never ever do this mistake again.

All the mistakes the dear reviewers have mentioned were completely correct and acceptable and I am very grateful that you read my article carefully.

Thank you for giving me another chance to send you the revised manuscript. I hope I can send you more articles and have more cooperation with you.

Kind regards

Golnaz Shambayati

---

## [Editor Report · Decision Letter 3]

25 Mar 2025

Production of probiotic garden cress (Lepidium Sativum) using Bifidobacterium Bifidum and its evaluation of nutritional value, biocontrol and growth rate ability

PONE-D-24-18993R3

Dear Dr. Shambayati,

We’re pleased to inform you that your manuscript has been judged scientifically suitable for publication and will be formally accepted for publication once it meets all outstanding technical requirements.

Kind regards,

Shengqian Sun

Academic Editor

PLOS ONE
---

## [Editor Report · Acceptance letter]

PONE-D-24-18993R3

PLOS ONE

Dear Dr. Mohammadzadeh Vazifeh,

I'm pleased to inform you that your manuscript has been deemed suitable for publication in PLOS ONE. Congratulations! Your manuscript is now being handed over to our production team.

Kind regards,

on behalf of

Dr. Shengqian Sun

Academic Editor

PLOS ONE